# Confocal Laser Scanning Microscopy of Light-Independent ROS in *Arabidopsis thaliana* (L.) Heynh. TROL-FNR Mutants

**DOI:** 10.3390/ijms26147000

**Published:** 2025-07-21

**Authors:** Ena Dumančić, Lea Vojta, Hrvoje Fulgosi

**Affiliations:** Laboratory for Molecular Plant Biology and Biotechnology, Division of Molecular Biology, Ruđer Bošković Institute, Bijenička Cesta 54, 10000 Zagreb, Croatia; edumanc@irb.hr (E.D.); lea.vojta@irb.hr (L.V.)

**Keywords:** abiotic stress, Arabidopsis, ferredoxin:NADP^+^, oxidoreductase, linear electron transport, redox homeostasis

## Abstract

Thylakoid rhodanese-like protein (TROL) serves as a thylakoid membrane hinge linking photosynthetic electron transport chain (PETC) complexes to nicotinamide adenine dinucleotide phosphate (NADPH) synthesis. TROL is the docking site for the flavoenzyme ferredoxin-NADP^+^ oxidoreductase (FNR). Our prior work indicates that the TROL-FNR complex maintains redox equilibrium in chloroplasts and systemically in plant cells. Improvement in the knowledge of redox regulation mechanisms is critical for engineering stress-tolerant plants in times of elevated global drought intensity. To further test this hypothesis and confirm our previous results, we monitored light-independent ROS propagation in the leaves of *Arabidopsis* wild type (WT), TROL knock-out (KO), and TROL ΔRHO (RHO-domain deletion mutant) mutant plants in situ by using confocal laser scanning microscopy with specific fluorescent probes for the three different ROS: O_2_^·−^, H_2_O_2_, and ^1^O_2_. Plants were grown under the conditions of normal substrate moisture and under drought stress conditions. Under the drought stress conditions, the TROL KO line showed ≈32% less O_2_^·−^ while the TROL ΔRHO line showed ≈49% less H_2_O_2_ in comparison with the WT. This research confirms the role of dynamical TROL-FNR complex formation in redox equilibrium maintenance by redirecting electrons in alternative sinks under stress and also points it out as promising target for stress-tolerant plant engineering.

## 1. Introduction

Photosynthesis is a very important biochemical process that is carried out by photoautotrophic organisms. They use the energy of photons, as well as water molecules as the electron donor and the source of protons, in order to fix carbon dioxide (CO_2_) from the atmosphere to finally produce simple sugars [1]. A very important part of the photosynthetic electron transfer chain (PETC) is the last reversible linear step of electron transfer from the small soluble protein ferredoxin (Fd) to the nicotinamide adenine dinucleotide phosphate (NADP^+^) [2]. This step is catalyzed by a dimerized flavoenzyme ferredoxin-NADP^+^ oxidoreductase (FNR) that can be attached to photosynthetic membranes via different proteins [3,4,5], but herein we focus on the interaction with thylakoid rhodanese-like protein (TROL) [6,7]. FNR bound to TROL efficiently catalyzes NADPH biosynthesis, while its soluble form prevents reactive oxygen species (ROS) formation and damage of molecular structures, suggesting the protective role of this complex [5,8]. Besides its fundamental role in sustaining life on Earth, photosynthesis is also the primary source of ROS production in plants. Also, as in other organisms, ROS are generated in various cellular compartments other than chloroplasts and have the ability to transition between them [9]. Because of this, in this study, we focused on ROS propagation and scavenging at the whole-cell level in leaves, including ROS not directly linked to the PETC.

TROL consists of two transmembrane domains that span through the thylakoid membrane, an inactive rhodanese domain (RHO) in the thylakoid lumen, TROL protein C-terminal domain (ITEP), that binds FNR in stroma, characteristic of vascular plants, and the N-terminal stromal domain with a chloroplast targeting pre-sequence [8,10]. Upstream of the ITEP domain, the TROL protein polyproline domain (PEPE) is located. This domain contains a characteristic proline–valine–proline repeat-rich region that enables flexibility [5]. TROL is located in the thylakoid membranes like a hinge between the end of the protein complexes of the PETC and the precursor of the Calvin cycle, NADP+ [10]. The inactive RHO domain differs from the active one by the aspartate residue replacing cysteine 207 in the active site. The inactive RHO domain of TROL protein might be responsible for the transmembrane signal transduction from lumen to stroma that influences TROL-FNR dynamical binding and, consequently, partitioning of photosynthetic electrons. It possibly causes conformational changes in the ITEP domain, which leads to the release of FNR when over-reduction of the PETC occurs [11]. It has been postulated that inactive RHO domains participate in cooperation with MAPK phosphatases (a large group of signaling, regulatory enzymes), thioredoxins (TRXs), and various stress response proteins. These domains are implicated in the maintenance of redox homeostasis and the detoxification of ROS across different intracellular processes. This implies a role for inactive RHO domains in signal transduction and cellular regulation. In *A. thaliana,* rhodanese activity is elevated under stress conditions, and stomata opening is at least partly regulated by receptors containing inactive RHO domains. Regulation of TROL-FNR binding is important for balancing the redox status of stroma with the membrane ETC. Such regulation is critical for preventing over-reduction in these compartments and maintaining redox poise. Under stress conditions, it is hypothesized that FNR release from TROL could be triggered by signals originating from the RHO domain upon interaction with specific signaling molecules, leading to conformational changes in the ITEP domain. NMR structural analysis of the TROL RHO domain has revealed a slightly altered loop that may accommodate plastoquinone (PQ) as a signal molecule. Additionally, progression of the eucaryotic cell cycle, controlled by Cdc25 phosphatase (active rhodanese domain), is influenced by intracellular redox changes. Quinone binding has been shown to modulate its activity, further highlighting the redox-sensitive regulatory functions of rhodanese domains [5,10].

In the PETC, photosystem I (PSI) and photosystem II (PSII) are the main sources of ROS propagation (singlet oxygen (^1^O_2_) and superoxide radical (O_2_^·−^)) (Figure 1), especially under stress conditions [12]. Besides chloroplasts, ROS are generated in the peroxisomes, mitochondria, cytosol, apoplast, endoplasmic reticulum, and cell wall [13]. It is well known that ROS in certain amounts serve as secondary messengers, but increased amounts can, due to their high reactivity, cause damage to DNA, RNA, lipids, proteins, and the whole cells. To prevent this critical damage, a balance between ROS propagation and scavenging needs to be maintained. This is one of the reasons why plants have developed various antioxidant defense mechanisms, since they are sessile organisms [14]. Electron excess in the PETC disrupts energy imbalance, manifesting as (1) LEF saturation via overoxidation or (2) ROS overproduction via over-reduction-induced photooxidation. This imbalance of energy does not remain limited just within chloroplasts; it disturbs the homeostasis of the whole plant [15,16]. Due to stress conditions, damage, and energy imbalance, linear electron transport (LET) components (e.g., PSI) cannot transfer electrons further in LET, so they become overoxidized on donor or acceptor sites and cause overoxidation on downstream elements, reducing the amounts of NADPH available for the Calvin cycle and ATP synthesis. On the other side, upstream elements become over-reduced [17]. At this point, dynamical binding of FNR to TROL and the consequent ability of this complex to regulate redox status come to focus [1,6,18]. Although ROS are short-lived and primarily generated in chloroplasts, they can travel between cellular compartments through membrane diffusion, aquaporins, and vesicle transport. Additionally, rather than physically moving between compartments, ROS can transmit redox signals and influence the antioxidant system by modifying oxidized glutathione, thiols, transcription factors, peroxiredoxins (PRXs), TRX, and glutaredoxins (GRXs). In addition to this, ROS propagation multiplies over time. Furthermore, mitochondria–chloroplast contact sites and peroxisome–mitochondria interactions create specialized microdomains that facilitate efficient ROS exchange, allowing them to affect different organelles without requiring long-distance diffusion [9,19]. It is also worth mentioning that FNR reductive activity, which ensures transfer of electrons from NADPH back to Fd, is important for handling part of the redox imbalance [20]. It has been proposed that when the NADP^+^/NADPH ratio changes and the PETC is over-reduced, FNR detaches from TROL due to conformational changes facilitated by the transmembrane signal, which is possibly mediated by the lumen-located RHO domain. Consequently, FNR starts to redirect the surplus of electrons to ROS scavengers and alternative electron sinks are activated (Figure 1) [8]. Besides electron transfer to NADP^+^, Fd, with the help of Fd-dependent thioredoxin reductase (FTR), transfers electrons to redox-regulating enzymes TRXs. In general, the equilibrium between ROS formation and scavenging is under the control of nonenzymatic (carotenoids, glutathione (GSH), and ascorbate (ASC)) and enzymatic antioxidants. Cu/Zn and Fe superoxide dismutases (SODs) are the first-line defense enzymatic antioxidants that act against superoxide anion by catalyzing its conversion to hydrogen peroxide (H_2_O_2_). H_2_O_2_ can cause the formation of another ROS, namely hydroxyl radical (OH∙) (Figure 1). The amounts of H_2_O_2_ are under the control of ascorbate peroxidases (APXs) and thiol-dependent peroxidases (TPXs). APXs use ascorbate as an electron donor for H_2_O_2_ reduction and the consequent formation of monodehydroascorbate (MDA), which is then oxidized to dehydroascorbate (DHA). MDA and DHA reductases use GSH as an electron donor to regenerate ASC, while regeneration of GSH is under the control of the NADPH-dependent GSH reductase (GR). In the TPX group, PRX and GSH peroxidases (GPXs) are included. Maintenance of chloroplast redox equilibrium is linked to the H_2_O_2_ scavenging system via the disulfide reductase activity of the NADPH-dependent redox system (NTRC), as well as TRX and GRX, which enable TPX activity. NADPH serves as an electron donor for H_2_O_2_ antioxidative enzymes, while on the other side, H_2_O_2_ serves as an electron sink, indicating the role of these molecules in the regulation of redox status through antioxidant systems [21].

Although TROL is part of the PETC within chloroplasts, we proposed its influence on electron management and the redox homeostasis system of whole leaf cells [22]. This role arises from the necessity of preventing ROS damage across all cellular compartments, particularly under stress conditions [23]. Since ROS can be transferred across various cell compartments, it is justified to wonder if the TROL-FNR complex could partly scavenge ROS that are not propagated in chloroplasts and, since chloroplasts are the main site of ROS production in plants, if managing ROS propagated in chloroplasts protects other organelles and cells from diffusion of chloroplast-generated ROS.

Exposure to light during sample preparation or imaging can artificially elevate ROS levels in plant tissues, particularly through enhanced ROS production in chloroplasts. These light-induced ROS can obscure treatment-related oxidative changes, such as those caused by drought stress and TROL mutations [24]. To minimize such artifacts, we used the pre-experiment dark incubation approach. A dark period helps stabilize endogenous ROS levels by reducing light-induced ROS production and allowing the antioxidant systems to re-equilibrate the cellular redox state [9,25]. This provides a more accurate baseline for detecting ROS specifically attributable to experimental treatments. In photosynthetic tissues, dark incubation halts light-driven electron transport, preventing transient ROS fluctuations and enhancing the reliability of fluorescence-based measurements [26]. Additionally, some ROS, such as H_2_O_2_, may accumulate in the absence of light due to ongoing metabolic processes like mitochondrial respiration or reduced scavenging. In these cases, dark incubation may increase the detectable signal in non-chloroplastic compartments, improving sensitivity without compromising specificity [27,28]. Dark incubation also protects ROS-sensitive fluorescent dyes. Many of these dyes are photosensitive and prone to photobleaching (permanent fluorescence loss due to prolonged light exposure, leading to reduced sensitivity and false negatives) [29,30], photooxidation (dye oxidation by light in the presence of oxygen, generating fluorescence signals unrelated to biological ROS, thus producing false positives) [31], and loss of specificity (light-induced degradation or side reactions that result in non-specific fluorescent products) [29]. Moreover, delayed luminescence from photosynthetic tissues (weak light emission following light exposure) can interfere with accurate ROS imaging. Dark incubation allows this luminescence to decay, further improving the signal-to-noise ratio [32,33,34]. Studies have demonstrated that fluorescence microscopy can induce ROS production and photobleaching in plant cells, affecting cell physiology and potentially confounding observations. Protocols for ROS detection consistently recommend minimizing light exposure during dye loading and imaging to reduce artifacts and improve the accuracy of ROS quantification. Maintaining samples in darkness after dye application helps prevent premature dye oxidation and preserves fluorescence intensity and specificity [35,36,37].

We live in times of huge global climate changes, including lack of rainfall and extremely high temperatures, resulting in severe drought periods. Besides this, the global population is constantly increasing, bringing along elevated requests for food; therefore plant cultivation and crop production must fulfill high demands. All these and many more other reasons elevate the importance of research with the goal of plant yield improvement and increased plant tolerance and resistance to stress [38]. Photosynthesis is one of the primary targets in research within this scope, since it is directly affected by changes in climate and is also proportional to crop productivity. Under conditions of stress, like drought, balance in the transport of electrons is disturbed and ROS are propagated and distributed across cells, with redox imbalance persisting even beyond photosynthesis processes and chloroplasts. Since drought disrupts CO_2_ intake, the consequences are over-reduction of the PETC and restrictions in plant growth and development, which is a significant agricultural obstruction. The goal of this research was to further test the hypothesis that the dynamic interaction of FNR with TROL is responsible for the activation of alternative electron sinks in vascular plant photosynthesis and consequent redox homeostasis maintenance in the whole plant. We also wanted to confirm our previous results obtained by electron paramagnetic resonance spectroscopy (EPR) measurements that indicate lower O_2_^·−^ levels in the TROL KO mutant line in comparison with WT [8]. Decrease in ROS in TROL mutant lines was once more successfully detected but with a different technique, this time by using an in situ ROS detection approach with confocal laser scanning microscopy in combination with fluorescent dyes. Because ROS are short-lived, every detection method has certain limitations. Moreover, detecting ROS at specific sites and under precise conditions while determining their origin remains a significant challenge [29]. To overcome this, we incubated plants in darkness before measurement and performed the experiment on the whole-cell level in leaves, since detection of ROS at specific sites is hard. This approach minimized light-induced ROS artifacts and overcame compartment-specific detection challenges. In addition, although the role of the TROL-FNR protein pair in chloroplast redox homeostasis is well understood, its influence on whole-cell redox homeostasis remains to be confirmed. Since electron transport processes are nearly inactive in the dark, this approach allowed us to examine ROS propagation, distribution, and redox state at the cellular level while minimizing dominant ROS production in the PETC. Additionally, dark incubation with ROS-detecting fluorescent probes helped eliminate potential ROS formation caused by the fluorescent probes themselves. Also, the laser excitation wavelength was in the green spectral region, which is the least efficient for driving photosynthesis and auto-producing ROS from fluorescent probes. Taken together, the results demonstrated how TROL and its various mutations influence the redox state at the whole-plant leaf level.

## 2. Results

In this work we examined the propagation and the distribution of the three common ROS, superoxide anion, hydrogen peroxide, and singlet oxygen, in the leaves of the model plant *Arabidopsis thaliana* (L.) Heyn. ecotype Columbia (Col-0) (WT) and the two mutant lines TROL KO and TROL ΔRHO under different growing conditions. Differences in the physico-chemical characteristics of substrates under different growing conditions are presented in Appendix A: Physico-chemical characteristics of growing substrate under normal moisture and drought stress. The main difference between substrates is water content, which under drought stress conditions is ≈23% lower in comparison with normal moisture conditions. Electric conductivity is also significantly lower (≈96%) under drought stress, while the pH of the substrate under drought stress is ≈8.4% higher. Other differences include the amount of dry and organic matter, salt content, and macronutrient compounds (Appendix A). ROS differences between mutants and WT under contrasting growth conditions are visualized in the green field images for O_2_^·−^ (Figure 2), H_2_O_2_ (Figure 3), and ^1^O_2_ (Figure 4). Noticeable differences in green color intensity refer to different amounts of ROS present. The green color intensity indicates signal strength created upon the reaction of ROS with fluorescent dye. Figure 5 (O_2_^·−^), Figure 6 (H_2_O_2_), and Figure 7 (^1^O_2_) represent quantified results.

Only significant reduction in the amount of O_2_^·−^ was noticed under the conditions of drought stress in the TROL KO mutant line in comparison with the WT (Figure 2 and Figure 5). According to the results presented, the examined protective results are not active under normal moisture conditions (Figure 2 and Figure 5).

Only significant reduction in the amount of H_2_O_2_ was noticed under drought stress in the TROL ΔRHO mutant line in comparison with the WT (Figure 3 and Figure 6). Under the conditions of normal moisture, there is no activation of protective mechanisms (Figure 3 and Figure 6).

For the ^1^O_2_, only significant reduction can be noticed under the conditions of normal growth substrate moisture, contrary to the previous results. The amount was significantly reduced in the TROL ΔRHO mutant line in comparison with the WT (Figure 4 and Figure 7). Neither of the mutations shows a protective role against ^1^O_2_ under the drought stress conditions (Figure 4 and Figure 7).

## 3. Discussion

A short dark incubation period (2 h) was applied uniformly to all plants prior to ROS imaging to improve data accuracy and reproducibility. While such a period may modestly influence ROS levels and antioxidant enzyme activity, its overall effect is context-dependent and influenced by prior environmental conditions, such as drought stress, which is the interest of this study [39,40]. Drought-stressed leaves frequently accumulate ROS during light exposure due to overexcitation of the photosystems and impaired electron transport, particularly in PSI and PSII. This can lead to the rapid propagation of ^1^O_2_ and O_2_^·−^ [41]. The 2 h dark period reduces these acute, light-driven ROS bursts, thereby enabling a more accurate assessment of endogenous, stress-induced ROS accumulation rather than transient photooxidative events. In plants previously exposed to oxidative stress, the dark period may also help partially reset redox homeostasis and stabilize ROS-sensitive signaling proteins [14]. It supports a more consistent baseline for comparing mutant lines even after recent photosynthetic activity. Additionally, the dark period has multiple technical benefits during ROS detection: it reduces background autofluorescence from photosynthetic tissues, prevents nonspecific dye oxidation during loading and incubation, minimizes photobleaching, preserving the fluorescence intensity and specificity of light-sensitive ROS probes, and improves the signal-to-noise ratio, enhancing detection sensitivity [29,30,31]. By applying the dark period uniformly across samples, we ensured that any influence from this incubation step was controlled and consistent, allowing for reliable comparisons between mutants and WT, also as treatments. The dark period is not long, so the consequences of ROS propagated in the light are still present. The redox state of the whole leaves is examined because it is important to obtain insight into redox state after the growth light conditions [42].

Plants confront many severe environmental conditions, but they are successful in overcoming them as they have evolved numerous short- and long-term adaptation and acclimation mechanisms. Dynamical binding and positioning of the photosynthetic complexes can change depending on growth conditions, especially stress conditions like drought [43]. One factor of drought stress is limitation of electron donation by reducing the photolysis of water, which impairs electron flow through the ETC. Even though drought limits water (the source of electrons), the bigger limitation is on the downstream consumption of those electrons, not necessarily their generation [44]. Stomata are closed so CO_2_ intake is lower and the Calvin cycle slows down, causing lower regeneration of NADP^+^. Light still excites PSI and PSII and electrons are passed down the chain but there is no NADP^+^ so electrons back up, which causes over-reduction of electron carriers [44,45,46,47]. Since LET cannot accept more electrons, they leak and ROS are propagated. Reduced PSs are subjected to photoinhibition [45,46,48,49,50,51,52]. This leads to ETC over-reduction, increasing excitation pressure and electron leakage. To dissipate excess electrons, photorespiration and the water cycle become crucial, and CEF and antioxidant enzymes are upregulated to maintain redox balance [45,47]. One of the mechanisms previously presented by our group is that soluble FNR handles oxidative stress better than when docked to TROL, consequently enabling TROL KO plants to propagate lower amounts of ROS. Different TROL mutations affect TROL-FNR dynamic binding, and some of the mutations cause more efficient ROS scavenging. The TROL-FNR pair regulates electron transport and dynamic partitioning in accord with the energy requirements, particularly NADPH synthesis, and perhaps ATP. Since LET products are used in various metabolic reactions, TROL-FNR regulation of electron transport consequently influences the whole organism [5,8,18,53]. It is not TROL that is directly involved in ROS scavenging and energy production, but the flavoenzyme FNR. TROL mutant lines are involved in FNR binding and release from the vicinity of PSI. It is also to be expected that the RHO domain of TROL has an influence on electron partitioning by possible transmembrane redox signal transduction from lumen to stroma with a consequent influence on the dynamical binding of the TROL-FNR complex. It appears that scavenging reactions and alternative electron partitioning, driven by changes in the dynamic binding of the TROL-FNR pair, influence ROS metabolism and the maintenance of redox homeostasis at the whole-leaf cell level [5,8]. We examined the influence of different soil moisture conditions on ROS accumulation. We further tested our hypothesis, which arises from the previous work in our laboratory carried out by Vojta et al. [8,22], which postulates that alternative pathways of electron partitioning (Figure 1) under different environmental conditions depend on TROL-FNR complex dynamical formation. We investigated ROS distribution under the influence of TROL, not just on the chloroplast level but on the whole-leaf tissue. In this work we used confocal laser scanning microcopy in combination with specific fluorescent probes for visualization of three different ROS [14].

Firstly, we established that O_2_^·−^ formation in the TROL KO mutant line was significantly reduced in comparison with WT under drought stress (Figure 2 and Figure 5), in accordance with previous EPR measurements [14], indicating the protective role of FNR when TROL is not present. In the absence of thylakoid membrane docking sites, soluble FNR participates in electron partitioning to alternative electron sinks and successful scavenging of propagated O_2_^·−^ (Figure 1). There is also less spillage of electrons to O_2_, as earlier proposed [8,22,54]. It is important to mention that under growth-light conditions (GL), the electron transport rate is not lower in TROL KO. In fact, under GL, TROL KO plants grow more successfully than the WT, which is consistent with earlier PAM measurements that showed that the electron transport rate is not lower in TROL KO [5]. TROL-FNR binding becomes linear electron transport rate-limiting at quite high photosynthetically active radiation (PAR), which causes the release of FNR from TROL. The enzyme FNR is involved in other electron transport reactions besides linear photosynthetic electron transport (Figure 1). In addition, it takes two reduced ferredoxin molecules to generate one NADPH. TROL docks FNR in the vicinity of photosystem I, so reduced ferredoxin can efficiently transfer electrons to NADP^+^. When TROL is not present, FNR is not efficiently docked to the thylakoid membranes and electrons are distributed in alternative pathways, which take electrons more rapidly (Figure 1) [8]. This, however, happens at very high light intensities, or when plants are exposed to more than one stressor (combined stress). Additionally, the reduction in the negative stress effects enabled by the TROL-FNR pair in the PETC seems to be so significant that it positively influences the maintenance of a balanced redox state across the entire cell. ROS are usual byproducts generated in various metabolic processes. Formation of ROS is caused by partial reduction or energy transfer to O_2_. The major site of O_2_^·−^ propagation by the excess electron spillage to O_2_ is PSI, in whose vicinity on the thylakoid membrane TROL is positioned (Figure 1). Such proximity of the O_2_^·−^ propagation site and the TROL-FNR complex further points toward the protective role hypothesis [8,22]. In addition to chloroplasts, ROS are generated in various cellular sites. These include mitochondria, particularly in the respiratory ETC under stress or when ATP synthesis is impaired, peroxisomes, especially under high light (HL) intensity when photorespiration is elevated, NADPH oxidases in the plasma membrane, peroxidases in the cell wall, xanthine oxidase reaction sites, and autoxidation of redox compounds under stress conditions [55,56,57]. These results confirm the protective role and scavenging activity dependent on the TROL-FNR complex dynamical binding that is activated under stress conditions when an alternative electron sink is preferred over the LEF, and they imply a positive influence on the whole-leaf level (Figure 1). These results indicate that ΔRHO mutation has no effect on O_2_^·−^ scavenging, at least not on such a high level as in TROL KO line. The results also indicate that under unstressed conditions there is no increased scavenging activity (Figure 2 and Figure 5).

With these results in mind, we wanted to investigate whether other ROS are also susceptible to scavenging influenced by the lack of TROL, especially since ROS can be transported in various cell compartments, be propagated by chain reaction, and initiate propagation of each other independently of their propagation mechanisms [58]. If ROS propagated in chloroplasts can migrate to other parts of the cell and vice versa then TROL should consequently affect the whole cell’s redox status. Under drought stress conditions TROL KO did not show significant scavenging of ROS in comparison with the WT. Contrary to this, ΔRHO showed a significantly lower amount of H_2_O_2_ (Figure 6). This result indicates reduced ROS scavenging efficiency, other than O_2_^·−^ in TROL KO plants, while other defensive mechanisms against H_2_O_2_ could be present and more expressed in the ΔRHO mutant line. The RHO domain was proposed to have a role in transmembrane redox signal transduction, which affects the dynamical binding of TROL and FNR and consequent flow of electrons [5]. Oxidative stress could regulate the release of FNR from thylakoids [18,59], and this regulation could be achieved through the RHO domain in combination with reduced PQ by redox sensing from the lumen through the thylakoid membrane, influencing TROL-FNR dynamic binding on the stromal side through conformational changes induced in the ITEP domain [11,60,61]. ΔRHO mutants efficiently reduce H_2_O_2_, potentially through enhanced TROL-tAPX complex formation [22], boosting antioxidant activity. In addition, as proposed in Vojta et al. [22], TROL most likely forms a complex with tAPX, which is an important antioxidant enzyme in the process of H_2_O_2_ scavenging. The existence of a tAPX isoform is proven in the vicinity of PSI, which is the place of H_2_O_2_ and O_2_^·−^ generation. ΔRHO mutation might also influence TROL-FNR complex interaction with tAPX [22]. Spectrophotometric measurement also showed increased APX activity in the TROL KO and ΔRHO mutant lines under drought conditions in comparison with WT (Dumančić and Fulgosi, unpublished data [62]). In plants, some ROS, especially H_2_O_2_, serve as secondary messengers in various signaling and gene expression pathways. In chloroplasts, H_2_O_2_ is mostly generated in PSII at a low rate, in the Mehler reaction, in photorespiration, and as a product of O_2_^·−^ reduction, with the latter also occurring in various cellular compartments [25,26,63]. Other sources of H_2_O_2_ production include mitochondria during aerobic respiration, peroxisomes during photorespiration and β-oxidation, oxidase enzymes in the cytoplasm, peroxidases in the cell wall, and auxin oxidation [63,64,65]. SOD is one of the first and most abundant defense antioxidant mechanisms against O_2_^·−^ [12]. Since SOD produces a great amount of H_2_O_2_ from superoxide, the protection against peroxide in cells is very high and efficient. According to this, Figure 6 shows efficient scavenging of peroxide under drought conditions.

Contrary to the TROL KO, Arabidopsis plants that overexpress TROL (TROL OX) failed to successfully cope with the ROS propagation, owing to the fact that more FNR was bound to the membrane in these plants [22]. The results presented in this work are in accordance with the mentioned results and also with the previously demonstrated successful O_2_^·−^ scavenging in TROL KO plants, even when they were treated with methyl-viologen (herbicide that strongly induces formation of O_2_^·−^), as shown by the EPR [8,22,66]. Considering all the mentioned findings, the proposal that TROL-FNR dynamic interaction has a significant role in maintaining redox homeostasis is even more strengthened (Figure 1). The positioning of TROL on the thylakoid membranes plays crucial role in controlling redox balance in chloroplasts, which is expected, as chloroplasts are the primary sites of ROS generation in plants. However, it seems that its significance extends beyond this. By regulating ROS equilibrium and electron partitioning, TROL could prevent redox imbalance affecting areas beyond the chloroplasts. This could be achieved by controlling and partitioning the electron flow, thereby reducing ROS formation and preventing subsequent chain reactions that lead to ROS multiplication and intercellular ROS transfer.

The formation of ^1^O_2_, which represents the first excited electronic state of O_2_, is mostly located in the vicinity of PSII, where interaction of O_2_ with triplet chlorophyll occurs. Additionally, by triggering lipid peroxidation, ^1^O_2_ can amplify its own production [67,68,69]. Certain peroxidases (PRX33, PRX34, APX, and guaiacol peroxidase (GPOX)) and lipoxygenases (LOX2, LOX3, and LOX4) can also generate ^1^O_2_ from fatty acids. And these reactions are not related only to chloroplasts [67,68]. ^1^O_2_ is a specific ROS because it is not generated by electron transfer to O_2_. The two main mechanisms for ^1^O_2_ scavenging are physical and chemical quenching [69]. According to the results of this research, neither of the tested mutant lines possess the capability of scavenging ^1^O_2_ under drought stress conditions (Figure 7).

Stress conditions such as drought interact with ongoing cellular signaling processes, often disrupting or modifying them. These processes rely on a wide array of signaling molecules whose synthesis, stability, and interactions are tightly interconnected and responsive to both internal and external cues. As a result, a cell’s physiological state reflects the integrated output of multiple overlapping and dynamic input signals. Rather than functioning through simple, linear pathways, plant cell signaling operates via a complex network with extensive crosstalk. One of the key intermediates in this network is ROS [70]. ROS are generated through various metabolic pathways and cellular compartments, and their presence influences numerous physiological processes. Their effects are context-dependent and determined by their localization, concentration, and duration of accumulation. ROS also serve as both intra- and intercellular messengers. They can move between organelles and cells, enabling coordinated communication during both normal metabolism and stress responses [9,70]. Under stress, such as drought, ROS production is often amplified, contributing to the propagation of stress signals. Importantly, different ROS types can influence each other’s formation, stability, bioavailability, and scavenging, further adding complexity to redox signaling [71,72,73,74]. Shifts in the cellular redox state caused by ROS can activate or suppress specific sets of redox-sensitive proteins, effectively functioning as molecular switches that allow cells to rapidly adapt to changing environmental conditions [70]. Maintaining redox homeostasis is essential for proper cellular function. Since ROS are inherently oxidizing, imbalances between their production and scavenging can disrupt the redox potential of the cell. This imbalance can affect multiple layers of cellular regulation, including signal transduction, transcriptome reprogramming, and transcription factor activity [75]. Ultimately, depending on the concentration and spatial distribution of ROS, outcomes can range from altered gene expression and protein function to broader physiological changes in the plant [76,77].

Knowledge gathered so far together with the results of this work indicates that formation of the TROL-FNR complex is triggered in situations where LEF and NADP^+^ synthesis is preferred, while when stress conditions are present, the release of FNR from TROL enables various alternative electron sinks and scavenging mechanisms in order to maintain whole-plant redox homeostasis. Exact mechanisms of ROS scavenging influenced by TROL-FNR complex dynamics remain to be determined. Further research should be carried out to answer these questions.

## 4. Materials and Methods

### 4.1. Plant Material and Growth Conditions

*Arabidopsis thaliana* (L.) Heynh. ecotype Columbia (Col-0) plants (originally obtained from the European Arabidopsis stock center, NASC, Loughborough, UK) were used as model organisms. Wild type (WT) and two mutant lines, TROL KO mutant line with the mutation on chromosome 4 in the gene *At4g01050* (T-DNA element SAIL_27_B04 insertion into the last intron at the position 2278 of *At4g01050*), which does not express TROL protein [5], and ΔRHO (13 amino acid (203–215) deletion in the RHO domain), were used [50,78]. Plants were grown from seeds in controlled-environment chambers. In total, 15 plants of each mutant line and the WT were grown in the growing system (Arasystem 3600 KIT, Betatech, Gent, Belgium); trays were divided into WT and two mutant lines. The substrate (A400, Stender, Schermbeck, Germany) was distributed in araflats (specially designed arrays consisting of 51 of individual pot cavities, perfectly suited for growing Arabidopsis plants at optimal densities) and soaked in trays overnight prior to sowing. Plant seeds were vernalized 2 days prior to sowing. Sowed seeds were covered with transparent foil until germination. Growth conditions were 21 °C, 440 ± 20 ppm CO_2_, and 12 h day/night photoperiod (equinox 21st May), under the LED illumination of CI-800 Programmable LED Experimentation System (CID Bio-Science, Inc., Camas, WA, USA), which mimics the sunlight on the geographic position of Birmingham city, West Midlands, UK (latitude 54.00, longitude −2.00). Plants were grown for 1 month under normal moisture conditions and then for 2 more weeks under arid conditions. The difference between humid and arid conditions was maintained by the gravimetric method, between 2.5 and 3.0 kg for humid and 2.0 and 2.4 kg for arid conditions. Plants were supplemented with ¼ Hoagland’s solution (1.25 mL/L Ca(NO_3_)_2_ × 4 H_2_O, 1.25 mL/L KNO_3_, 0.25 mL/L KH_2_PO_4_, 0.5 mL/L MgSO_4_ × 7 H_2_O, 0.25 mL/L micronutrients (2.86 g/L H_3_BO_3_, 1.81 g/L MnCl_2_ × 4 H_2_O, 0.22 g/L ZnSO_4_ × 7 H_2_O, 0.08 g/L CuSO_4_ × 5 H_2_O, 0.02 g/L NaMoO_4_ × H_2_O g/L), and 0.25 mL/L Fe-EDTA (10.4 g/L EDTA, 7.8 g/L FeSO_4_ × 7 H_2_O, 56.1 g/L KOH g/L)), pH 5.5 [79], every two weeks. Substrate chemical composition was determined and checked by the accredited analytical laboratory of the Department for Plant Nutrition, Division of Agroecology, Faculty of Agriculture, University of Zagreb, Svetošimunska cesta 25, 10,000 Zagreb, Croatia. Chemical composition analyses revealed subtle changes in the macronutrient composition between the different substrate lots. Those differences affected reproducible plant growth and development. To offset those differences, we supplemented both substrate types with chemically well-defined Hoagland’s nutrient solution. Together with that, reproducible growth conditions (light, temperature, CO_2_ level) enabled detection of changes in the amount of ROS in different TROL mutant lines under different growth conditions that are a consequence of the subtle differences in biochemical processes, morphology, and photosynthesis, which was the goal of this research.

### 4.2. Chemicals

Fluorescent probes dyhidroethidium (DHE) and Singlet oxygen sensor green (SOSG) were purchased from Thermo Fisher Scientific (Waltham, MA, USA) and Spy-LHP from Dojindo Molecular Technologies Inc. (Rockville, MD, USA).

### 4.3. Sample Preparation

Plants were 4 weeks old for the normal moisture conditions and 6 weeks old for the drought stress conditions. Morphologically, under normal moisture conditions leaves were of healthy green color with bigger rosettes in mutant lines than in WT. Under drought stress TROL KO plants had bigger rosettes of green color, while WT displayed a reddish color from accumulated anthocyanins. Morphological characteristics of TROL ΔRHO were in between TROL KO and WT. All sampled leaves were at the fully developed stage. Plants of each line were put in the dark 2 h before the experiment. The whole leaf was cut out from the plant and rinsed in 50 mM HEPES buffer (pH 7.5) and after that incubated for 30 min in the desired fluorescent probe (250 µM DHE, 50 µM Spy-LHP, or 50 µM SOSG), while control samples were incubated in 50 mM HEPES buffer. After incubation, the leaf was again rinsed in 50 mM HEPES buffer and transferred onto a glass slide in a drop of 50 mM HEPES buffer and covered with a cover glass. The protocol was modified according to Prasad et al. [14].

### 4.4. Confocal Microscopy

The amount and arrangement of ROS within leaves were visualized using a laser scanning confocal microscope TCS SP8 (Leica Microsystems GmbH, Wetzlar, Germany) (Figure 2, Figure 3 and Figure 4). The excitation for SOSG was 504 nm and emission 520–560 nm, for the Spy-LHP excitation was 524 nm, emission 535–580 nm, and for the DHE excitation was 480 nm, emission 560–610 nm. Chloroplast autofluorescence was determined at 650–750 nm for all fluorescent probes. At the beginning of each experiment, proper laser intensity was set by using control samples in which leaves were incubated in 50 mM HEPES buffer without fluorescent probes. All confocal microscopy experiments include images of chloroplast autofluorescence (red field signal), bright field images (gray field signal), and detected ROS with fluorescent probes (green field signal) (Figure 2, Figure 3 and Figure 4). For each fluorescent probe and control, 3 visual fields at different plane levels were recorded, yielding a sum of 10 images for each of the three leaf replicas. The experiment was carried out in situ.

### 4.5. Image Analysis

Images obtained with confocal laser scanning microscopy (Figure 2, Figure 3 and Figure 4) were analyzed by using ImageJ (*Fiji*) software (Java 1.8.0._345 (64 bit)). The green field fluorescent signal strength of fluorescent probe reactions with ROS was measured in total counts. Measurement was applied on the whole image. The average value of control images was subtracted from the average value of respective reaction groups to evaluate signal intensity. Ten measurements of each fluorescent probe and the control sample for every growth condition and plant line were used on the three leaf replicas. On the images represented in this paper, the contrast of the green field was enhanced equally across the whole image in favor of better visibility by GIMP 2.10.38. Measurements were performed on original, unprocessed images. Images of all 3 scanned fields and plant lines under the examined growth conditions were grouped for every detected ROS separately, in favor of easier comparison of the detected signals.

### 4.6. Statistical Analysis

All the data were expressed as mean ± standard deviation. Statistical analysis was carried out with GraphPad Prism (v.9.0.0.121). Comparison of groups was performed by using one-way ANOVA with post hoc analysis Dunnett′s test. The statistical significance is indicated as * at a *p* < 0.05 confidence level. Biological replicate = leaf from independently grown and randomly selected plants (*n* = 3 plants/plant line). Technical replicate = every leaf was imaged at 3 different locations at different plane levels for a total count of 10 images.

## 5. Conclusions

When considering improvement in plant/crop stress tolerance and protection, the TROL-FNR complex must inevitably be taken into consideration because of various dynamical interactions through which it can regulate electron transport in different pathways of the PETC and ROS scavenging in chloroplasts and beyond. The exact mechanisms of ROS scavenging pathways that involve the TROL-FNR complex remain to be elucidated. To conclude, this research brings additional evidence for the involvement of TROL-FNR interaction in ROS protection under stress conditions. Once more, we showed that dynamical binding of FNR to TROL significantly participates in the maintenance of redox equilibrium by activating various detoxication mechanisms against ROS (precisely O_2_^·−^ and H_2_O_2_) formed under, in this case, drought stress conditions. While confocal imaging captured spatial ROS patterns, biochemical assays are needed to confirm scavenging kinetics. In TROL protein, agricultural improvements can find a promising manipulation tool for improving crop resilience after validation in natural conditions.

## Figures and Tables

**Figure 1 ijms-26-07000-f001:**
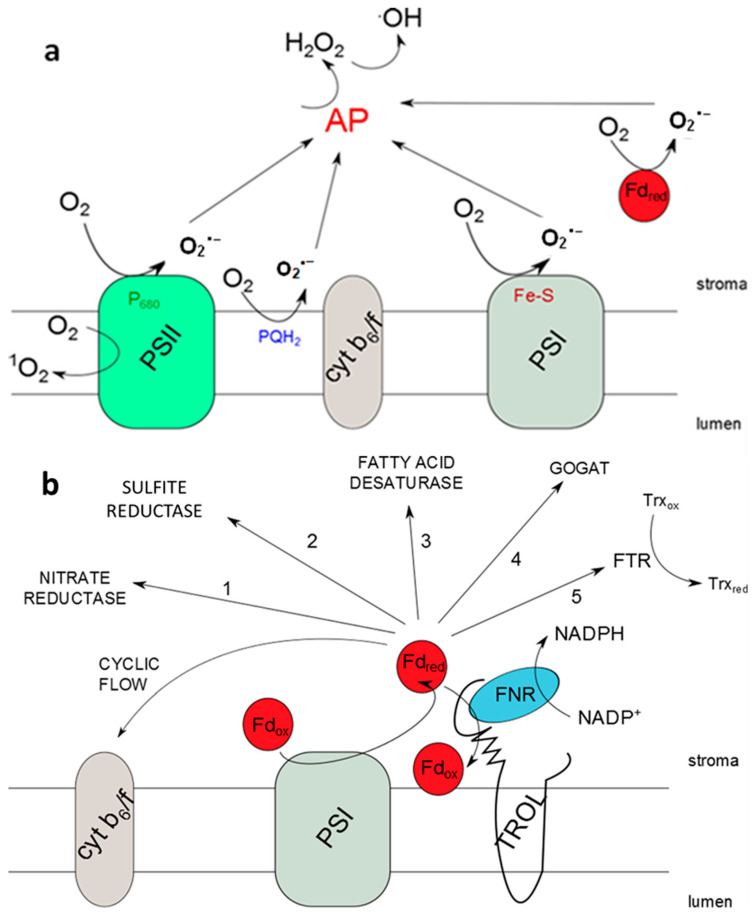
Photosynthetic reactive oxygen species (ROS) propagation, possible electron transfer pathways, and the role of the thylakoid rhodanese-like protein and ferredoxin-NADP^+^ oxidoreductase (TROL-FNR) pair. (**a**) ROS propagation sites in the photosynthetic electron transport chain by excitation and reduction of O_2_; (**b**) existing electron transfer pathways from reduced ferredoxin (Fd_red_) in the vicinity of photosystem I (PSI). When FNR is docked to thylakoid membranes by TROL, the linear electron transport pathway is preferred. In the absence of FNR binding to TROL, e.g., in the TROL knock-out (TROL KO) plants, electrons from Fd_red_ can be distributed to some of the alternative pathways. This transfer is more rapid than the linear electron transport pathway, efficiently preventing electron transfer to O_2_ and consequently causing lower superoxide anion (O_2_^·−^) propagation. Dynamical binding of the TROL-FNR protein pair influences the distribution of electrons in different pathways depending on energy status and needs of the cell.

**Figure 2 ijms-26-07000-f002:**
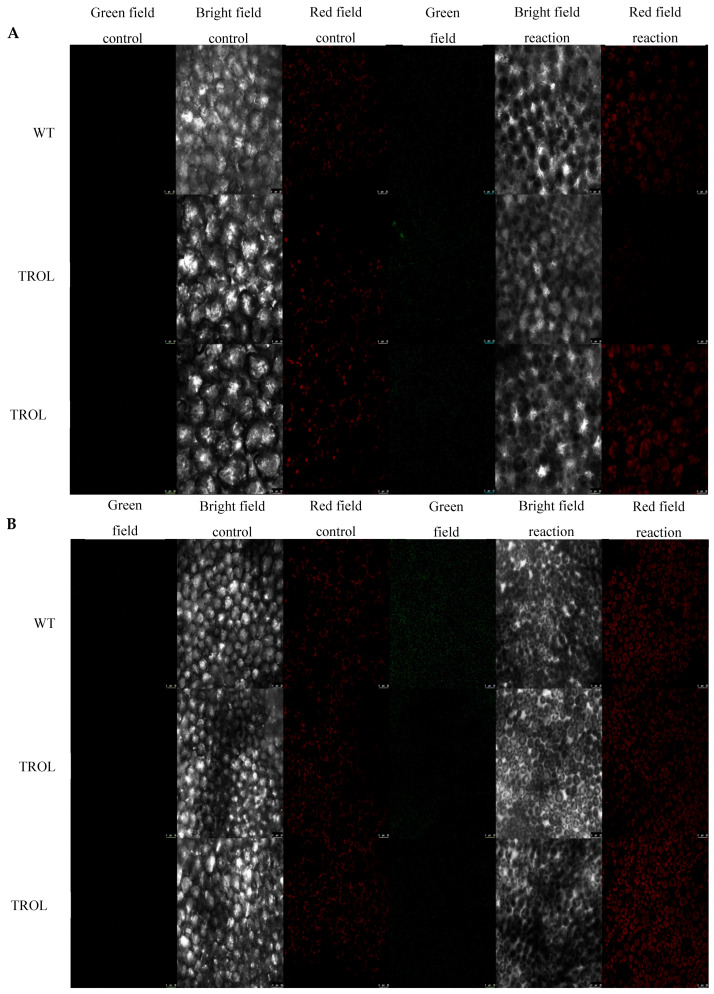
Confocal laser scanning microscopy images of superoxide radical (dihydroethidium, DHE). The gray field represents the non-fluorescent/background region of the sample, the red field indicates excitation of the fluorescent molecule and light emission at the particular wavelength, and the green field represents the fluorescent green signal created upon the reaction of DHE oxidation by O_2_^·−^ to form 2-hydroxyethidium. (**A**) Under normal moisture growth conditions; (**B**) under drought growth conditions. Samples are visualized under the same settings to enable comparison: HC PL APO CS2 63x/1.40 oil and identical camera settings. Representative images are shown. The contrast of green field images was enhanced in favor of better visibility and equal results interpretation across the whole image by GIMP 2.10.38. Measurements were carried out on original, unprocessed images. Images are grouped in favor of easier comparison of the detected signals. Scale bars: 25 μm.

**Figure 3 ijms-26-07000-f003:**
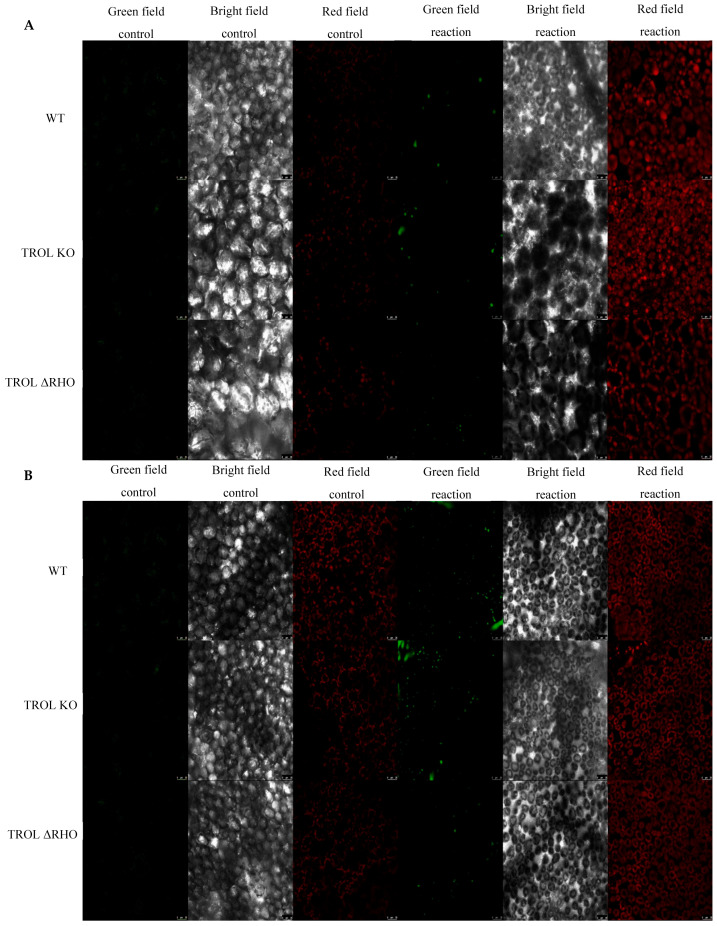
Confocal laser scanning microscopy images of hydrogen peroxide (Spy-LHP). The gray field represents the non-fluorescent/background region of the sample, the red field indicates excitation of the fluorescent molecule and light emission at the particular wavelength, and the green field represents the increased fluorescence green signal upon the oxidation reaction of Spy-LHP by H_2_O_2_. (**A**) Under normal moisture growth conditions; (**B**) under drought growth conditions. Samples are visualized under the same settings to enable comparison: HC PL APO CS2 63x/1.40 oil and identical camera settings. Representative images are shown. The contrast of green field images was enhanced in favor of better visibility and equal results interpretation across the whole image by GIMP 2.10.38. Measurements were carried out on original, unprocessed images. Images are grouped in favor of easier comparison of the detected signals. Scale bars: 25 μm.

**Figure 4 ijms-26-07000-f004:**
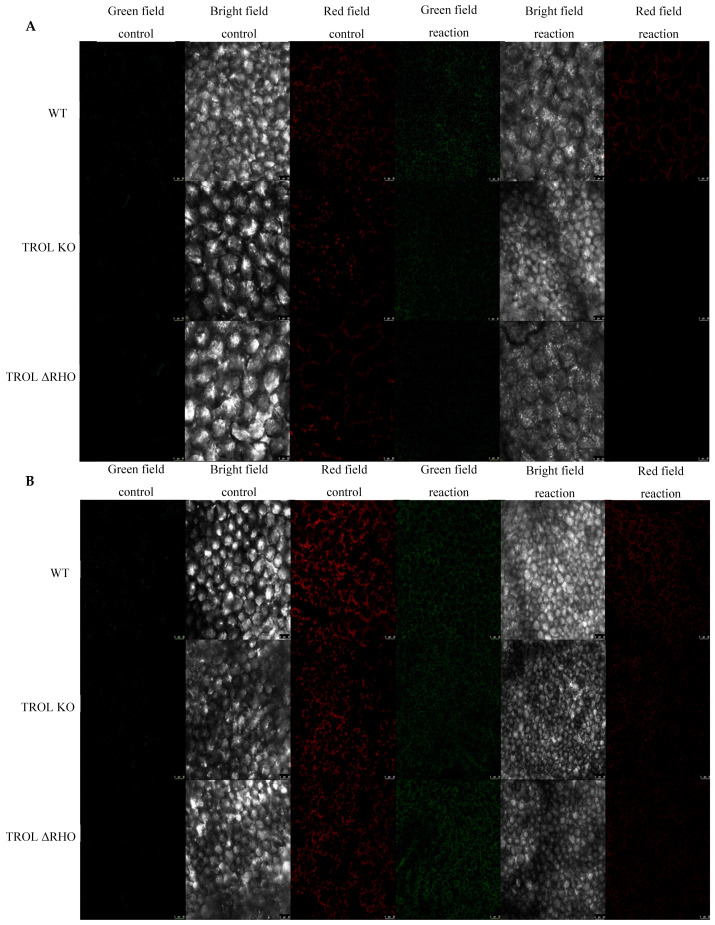
Confocal laser scanning microscopy images of singlet oxygen (singlet oxygen sensor green, SOSG). The gray field represents the non-fluorescent/background region of the sample, the red field indicates excitation of the fluorescent molecule and light emission at the particular wavelength, and the green field represents the fluorescence green signal that SOSG emits in the presence of ^1^O_2_. (**A**) Under normal moisture growth conditions; (**B**) under drought growth conditions. Samples are visualized under the same settings to enable comparison: HC PL APO CS2 63x/1.40 oil and identical camera settings. Representative images are shown. The contrast of green field images was enhanced in favor of better visibility and equal results interpretation across the whole image by GIMP 2.10.38. Measurements were carried out on original, unprocessed images. Images are grouped in favor of easier comparison of the detected signals. Scale bars: 25 μm.

**Figure 5 ijms-26-07000-f005:**
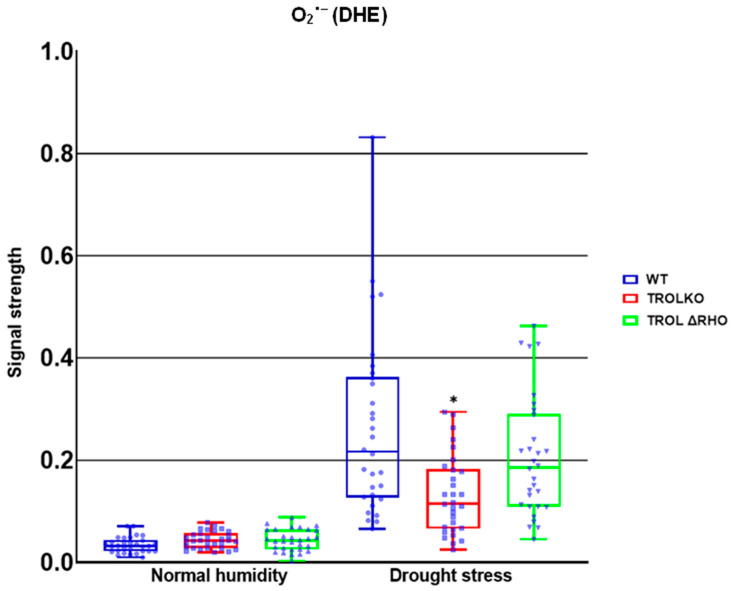
Confocal laser scanning microscopy signal strength detection results for superoxide anion. The distribution of all obtained results is shown. The statistically significant decrease in the superoxide anion (O_2_^·−^) amount in the TROL knock-out (TROL KO) mutant line under the drought stress condition indicates the protective role of the thylakoid rhodanese-like protein and ferredoxin-NADP^+^ oxidoreductase (TROL-FNR) complex dynamical binding and alternative electron sink formation. Statistical significance indicated as * at a *p* < 0.05 confidence level.

**Figure 6 ijms-26-07000-f006:**
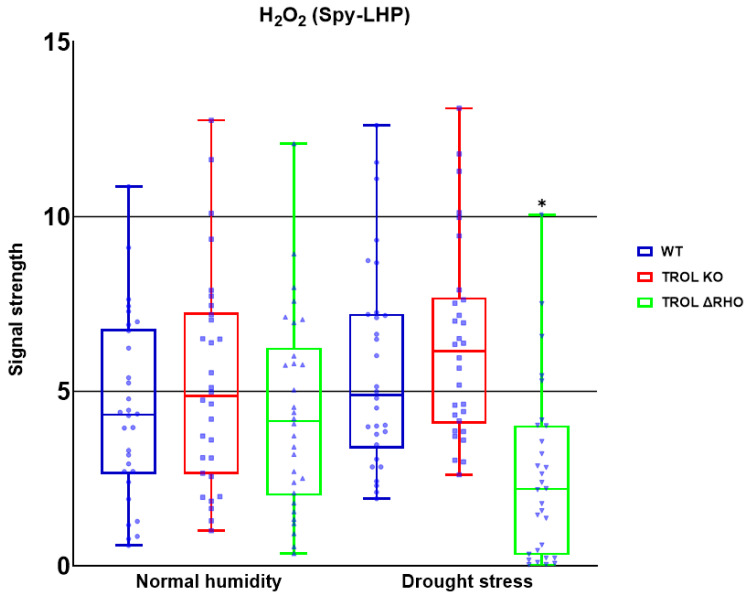
Confocal laser scanning microscopy signal strength detection results for hydrogen peroxide. The distribution of all obtained results is shown. The statistically significant decrease in the H_2_O_2_ amount in the ΔRHO mutant line under the drought stress condition indicates the defensive role of the TROL ΔRHO domain against H_2_O_2_ by transmembrane signal transduction, which influences the thylakoid rhodanese-like protein and ferredoxin-NADP+ oxidoreductase (TROL-FNR) complex dynamical binding and TROL complex formation with tAPX. Statistical significance indicated as * at a *p* < 0.05 confidence level.

**Figure 7 ijms-26-07000-f007:**
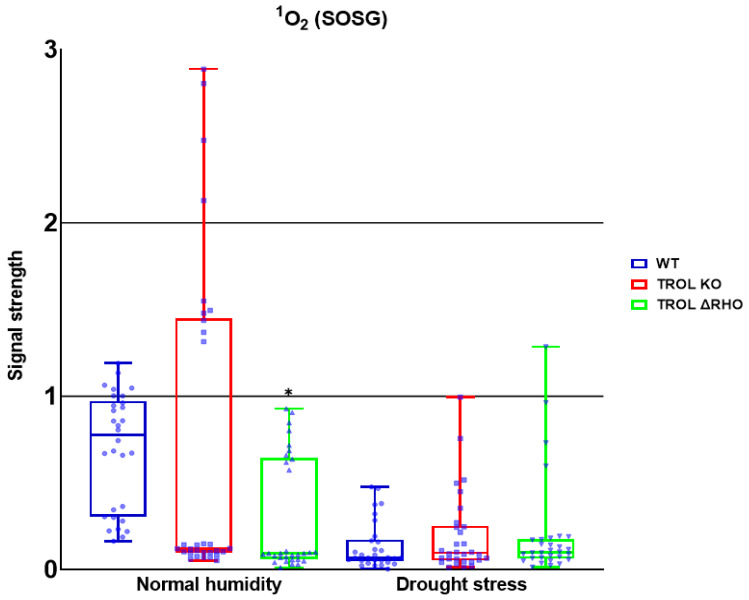
Confocal laser scanning microscopy signal strength detection results for singlet oxygen (^1^O_2_). The distribution of all obtained results is shown. A statistically significant decrease in the ^1^O_2_ amount in the ΔRHO mutant line under normal moisture conditions is observed. Statistical significance indicated as * at a *p* < 0.05 confidence level.

## Data Availability

Data are contained within the article and Appendix A. The original supporting data available in the *Figshare* repository: https://doi.org/10.6084/m9.figshare.25736829.v3, https://doi.org/10.6084/m9.figshare.29420834.v1, https://doi.org/10.6084/m9.figshare.29423183.v1, https://doi.org/10.6084/m9.figshare.29423228.v1, https://doi.org/10.6084/m9.figshare.29423252.v1, https://doi.org/10.6084/m9.figshare.29423306.v1, and https://doi.org/10.6084/m9.figshare.29423381.v1.

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
