# Peer review of "Confocal Laser Scanning Microscopy of Light-Independent ROS in Arabidopsis thaliana (L.) Heynh. TROL-FNR Mutants"

_ijms, 2025, doi:10.3390/ijms26147000_

Round 1

Reviewer 1 Report

Comments and Suggestions for Authors

Dear Authors!

The article is devoted to a very narrow, specific topic. It is not very clear to an unprepared scientist without additional reading of the literature in this topic.

The cited references are mostly out of the last 5 years (78%), probably due to very specific theme of investigation. I think the references are relevant.

The manuscript is scientifically sound and the experimental design is appropriate to test the hypothesis.

The manuscript’s results are reproducible based on the details given in the methods section.

The figures and images are appropriate. They are properly show the data. They are easy to interpret and understand. The data are interpreted appropriately and consistently throughout the manuscript. The conclusions are consistent with the evidence and arguments presented. I didn`t find any ethical problems.

My specific comments:

Decipher for the first time LET (Line 74), PRX, TRX, and GRX (Line 84),  DHE (Line 233), SOSG (Line 304) , Spy-LHP (Line 280), KO (Line 386)?

Line 235. You should add additional information about reaction. Is it the fluorescence of stained ROS?

You should add a list of peroxidases and lipoxygenases (Lines 467-468).

Comments on the Quality of English Language

Dear Authors!

Check for typos in the text. You should correct trough to through (Lines 46, 112), photosysrem (L131), spectrofotometrical to spectrophotometric (Line 437), dana to data (Line 439).

Rephrase the sentences on Lines 195-196, 383-384, 388-389.  They are not understandable. Change the sentence “This could be achieved by controlling electron flow and partitioning…” (Line 462) to “This could be achieved by controlling and partitioning the electron flow…”.  Change the sentence “Confocal  microscope Leica TCS SP8 Laser Scanning Confocal Microscope (Leica Microsystems GmbH, Germany)” to “Laser Scanning Confocal Microscope TCS SP8 (Leica Microsystems GmbH, Germany)” (Line 558-559).

Rephrase the sentence “Together with that, reproducible growth conditions (light, temperature, CO2 level), enabled detection of the subtle differences in biochemical processes, morphology, and photosynthesis of different TROL mutant lines, grown under different substrate moisture conditions, which was the goal of this research.” (Lines 538-541)”. You only estimated the amount of ROS, which reflects of the process of photosynthesis.

Author Response

Dear Special issue editors Dr. Agnieszka Hanaka and Dr. Sylwia Zielińska, Reviewers, and Editorial Office Members

We are grateful for the editors and reviewers fast response, time and effort invested. Reviewers comments were reasonable and relevant, therefore we agreed and revised manuscript according to them. Revision improved the content and clarified ambiguities of our submitted manuscript entitled "Confocal laser scanning microscopy of light-independent ROS in Arabidopsis thaliana (L.) Heynh. TROL-FNR mutants" by Dumančić et al. (ijms-3754836).

According to the Reviewers comments we improved English language. The sentences that Reviewers pointed out as unclear, are rewritten to be understandable. We added more informations about TROL protein RHO domain influence on TROL-FNR interaction in dependance of redox status. Reactions of fluorescent dyes with ROS are also more clarified. Differences in growing substrates under normal humidity and drought stress conditions are stated in the Supplementary table 1. Morphological characteristics and developmental stages of experimental plants are clarified, also as the way biological and technical replicates were used. The importance of this research in context of global warming and possible agricultural improvements is more pointed out, together with outcoming advances in the field, possible limitations of research and unresolved questions. Missing references are added, and reference list is updated according to the Journal requirements. Overall, respected Reviewers comments are gratefully accepted and implemented into the manuscript that can be found in the attachment, together with additional Supplementary table 1.

Here are also point by point responses to the Reviewers comments and their position in the manuscript.

Comments 1: Decipher for the first time LET (Line 74), PRX, TRX, and GRX (Line 84),  DHE (Line 233), SOSG (Line 304) , Spy-LHP (Line 280), KO (Line 386)?

Response 1: The abbreviations are now stated with the full name at the first mention. Exception is Spy-LHP because in our knowledge that is the name of fluorescent dye, not abbreviation. Only more detailed description would be full IUPAC name (2-(4-Diphenylphosphanylphenyl)-9-(1-hexylheptyl)anthra[2,1,9-def,6,5,10-d'e'f']diisoquinoline-1,3,8,10-tetraone), in revised manuscript: (line 300, page 12, Results). LET, Linear electron transport (line 97, page 3, Introduction), PRX, peroxiredoxins, GRX, glutaredoxins (line 107, page 3, Introduction), TRX, thioredoxins (line 66, page 2, Introduction), DHE, Dihydroxyethidium (line 255, page 10, Results), SOSG, Singlet oxygen sensor green (line 325, page 14, Results), KO, TROL KO (line 408, page 16, Discussion).

Comments 2: Line 235. You should add additional information about reaction. Is it the fluorescence of stained ROS?

Response 2: We added description of fluorescent signal that appears in the reaction of fluorescent dyes and ROS. In revised manuscript: fluorescent green signal created upon the reaction of DHE oxidation by O2˙ˉ to form 2-hydroxyethidium (line 257, page 10, Results), increased fluorescence green signal upon the oxidation reaction of Spy-LHP by H2O2 (line 302, page 12, Results), fluorescence green signal that SOSG emits in the presence of 1O2 (line 327, page 14, Results). 

Comments 3: You should add a list of peroxidases and lipoxygenases (Lines 467-468).

Response 3: We added the list of peroxidases PRX33, PRX34, APX, guaiacol peroxidase (GPOX) and lipoxygenases LOX2, LOX3, LOX4 (lines 493-494, page18, Discussion).

Comments 4: Check for typos in the text. You should correct trough to through (Lines 46, 112), photosysrem (L131), spectrofotometrical to spectrophotometric (Line 437), dana to data (Line 439).

Response 4: The typos are corrected, in revised manuscript: Through (line 51, page 2,line 136, page 4, Introduction, lines 453-454, page 17, Discussion), photosystem (line 146, page 5, Introduction), spectrophotometric (line 461, page 17, Discussion), data (line 464, page 17, Discussion).

Comments 5: Rephrase the sentences on Lines 195-196, 383-384, 388-389.  They are not understandable. Change the sentence “This could be achieved by controlling electron flow and partitioning…” (Line 462) to “This could be achieved by controlling and partitioning the electron flow…”.  Change the sentence “Confocal  microscope Leica TCS SP8 Laser Scanning Confocal Microscope (Leica Microsystems GmbH, Germany)” to “Laser Scanning Confocal Microscope TCS SP8 (Leica Microsystems GmbH, Germany)” (Line 558-559).

Response 5: Sentences on stated lines are rephrased in revised manuscript: Decrease of ROS in TROL mutant lines was once more successfully detected but with different technique, this time by using in situ ROS detection approach with the confocal laser scanning microscopy in combination with fluorescent dyes. (lines 208-211, page 7, Introduction). In the absence of thylakoid membrane docking sites, soluble FNR participates in electron partitioning to alternative electron sinks and successful scavenging of propagated O2˙ˉ (Fig. 1). There is also less spillage of electrons to O2, as earlier proposed (lines 404-407, page 16, Discussion). TROL-FNR binding becomes linear electron transport rate limited at quite high photosynthetically active radiation (PAR) which causes release of FNR from TROL. (lines 410-412, page 16, Discussion). This could be achieved by controlling and partitioning the electron flow… is changed (line 488, page 18, Discussion), also as Laser Scanning Confocal Microscope TCS SP8 (Leica Microsystems GmbH, Germany) (lines 593-594, page 20, Materials and methods). 

Comments 6: Rephrase the sentence “Together with that, reproducible growth conditions (light, temperature, CO2 level), enabled detection of the subtle differences in biochemical processes, morphology, and photosynthesis of different TROL mutant lines, grown under different substrate moisture conditions, which was the goal of this research.” (Lines 538-541)”. You only estimated the amount of ROS, which reflects of the process of photosynthesis.

Response 6: Sentence is rephrased and in revised manuscript is (lines 567-571, page 2, Materials and methods) Together with that, reproducible growth conditions (light, temperature, CO2 level), enabled detection of changes in the amount of ROS in different TROL mutant lines under different growth conditions that are consequence of the subtle differences in biochemical processes, morphology, and photosynthesis, which was the goal of this research. 

Reviewer 2 Report

Comments and Suggestions for Authors

Manuscript ID: ijms-3754836

Manuscript Title: Confocal laser scanning microscopy of light-independent ROS in Arabidopsis thaliana (L.) Heynh. TROL-FNR mutants

Journal Name: International Journal of Molecular Sciences

Reviewer Comments

Subject and title

The topic of the manuscript is relevant and interesting for the journal audience.

Abstract

Line 10: Rewrite: “Thylakoid rhodanase-like protein (TROL) serves as a thylakoid membrane hinge linking photosynthetic electron transport chain (PETC) complexes to NADPH synthesis”

Line 10-15: Add a sentence linking TROL-FNR’s role to climate resilience, e.g., "As drought stress intensifies globally, understanding redox regulation mechanisms is critical for engineering stress-tolerant crops."

Line 13-14: “Our prior work indicates the TROL-FNR complex maintains redox equilibrium in chloroplasts and systemically in plant cells.”

Line 17: ARHO is used without defining it as an RHO-domain deletion mutant. Correct to "TROL ARHO (RHO-domain deletion mutant)" at first mention.

Line 20: Replace vague descriptors (e.g., "successful scavenging") with quantitative trends: "TROL KO reduced O₂˙ˉ by …%, while ARHO reduced H₂O₂ by …%"

Line 23: Explicitly state how findings advance the field: e.g., "This confirms TROL-FNR’s role as a master switch redirecting electrons to alternative sinks under stress, offering targets for improving crop resilience"

Introduction

The introduction describes what the author hoped to achieve and clarifies the problem that is being investigated. However, the following points should be addressed:

Line 54-59: Expand on RHO’s redox-sensing mechanism: "Inactive RHO may relay luminal redox cues via conformational changes in ITEP."

Line 68-71: Rewrite: “Electron excess in PETC disrupts energy balance, manifesting as: (1) LEF saturation via overoxidation, or (2) ROS overproduction via overreduction-induced photooxidation.”

Line 189: Connect to climate change: “Drought disrupts CO₂ intake, causing PETC overreduction—a key agricultural challenge”

Line 198: Add references

Line 199-211: Explicitly state the unresolved question: "While TROL-FNR’s role in chloroplast redox balance is established, its impact on whole-cell ROS propagation remains unproven."

Line 201-202: Rewrite: “This approach minimized light-induced ROS artifacts and overcame compartment-specific detection challenges.”

Results

Incomplete drought characterization. No physiological data (e.g., Relative water content (RWC), stomatal conductance, proline content, relevant enzymes….) confirming drought stress severity. Add supplementary table with stress parameters.

Line 217-219: Rewrite: “ROS differences between mutants and WT under contrasting growth conditions are visualized in the green fields images (Figs. 2,4,6).”

Discussion

The following comments should be addressed:

Line 321: It's clear the authors moved the Materials and Methods section after the Discussion section and forgot to change the citation number. I think the numbering here should start at 39.

Line 348-349, 361, 441, 466, 467, 479: Add references.

Line 372, 433: Add the citation number: ‘Vojta et al. [… …]’

Line 432: Rewrite: “ΔRHO mutants efficiently reduce H₂O₂, potentially through enhanced TROL-tAPX complex formation [Ref. no. …], boosting antioxidant activity.”

Materials and Methods

The design of the experiment is appropriate for the purpose of the study. However, there are details that need to be added to clarify the experiment:

Line 512: The numbering of citations here should change according to the change that will be made in the Discussion section.

Line 512-513: Rewrite: “Plants were grown from seeds in controlled-environment chambers.”

Line 531: Nutrient solution pH affects ROS stability. Add Hoagland’s solution pH value.

Line 549: Mention the age of the plants and their morphological characteristics at that stage. Specify leaf developmental stage sampled.

Line 555: Correct: “… according to Prasad et al. [14].

Line 588: Address biological replicates: e.g. "Biological replicate = leaf from independently grown plant (n=… plants/genotype or line)."

Line 591-593: Delete: “5. Conclusions” and “This section is not mandatory ….”

Conclusion

The conclusion demonstrates the scientific value added to the research.

Line 604: Acknowledge limitations: Add: "While confocal imaging captured spatial ROS patterns, biochemical assays are needed to confirm scavenging kinetics.".

Line 604: Reframe agricultural implications: e.g., "TROL manipulation shows potential for enhancing crop resilience pending field validation."

References

The reference list must be formatted according to the journal's requirements.

Author Response

Dear Special issue editors Dr. Agnieszka Hanaka and Dr. Sylwia Zielińska, Reviewers, and Editorial Office Members

We are grateful for the editors and reviewers fast response, time and effort invested. Reviewers comments were reasonable and relevant, therefore we agreed and revised manuscript according to them. Revision improved the content and clarified ambiguities of our submitted manuscript entitled "Confocal laser scanning microscopy of light-independent ROS in Arabidopsis thaliana (L.) Heynh. TROL-FNR mutants" by Dumančić et al. (ijms-3754836).

According to the Reviewers comments we improved English language. The sentences that Reviewers pointed out as unclear, are rewritten to be understandable. We added more informations about TROL protein RHO domain influence on TROL-FNR interaction in dependance of redox status. Reactions of fluorescent dyes with ROS are also more clarified. Differences in growing substrates under normal humidity and drought stress conditions are stated in the Supplementary table 1. Morphological characteristics and developmental stages of experimental plants are clarified, also as the way biological and technical replicates were used. The importance of this research in context of global warming and possible agricultural improvements is more pointed out, together with outcoming advances in the field, possible limitations of research and unresolved questions. Missing references are added, and reference list is updated according to the Journal requirements. Overall, respected Reviewers comments are gratefully accepted and implemented into the manuscript that can be found in the attachment, together with additional Supplementary table 1.

Here are also point by point responses to the Reviewers comments and their position in the manuscript.

Comments 1: Line 10: Rewrite: “Thylakoid rhodanase-like protein (TROL) serves as a thylakoid membrane hinge linking photosynthetic electron transport chain (PETC) complexes to NADPH synthesis”

Respond 1: Sentence is rewritten and in revised manuscript is (lines 10-12, page 1, Abstract) Thylakoid rhodanase-like protein (TROL) serves as a thylakoid membrane hinge linking photosynthetic electron transport chain (PETC)  complexes to nicotinamide adenine dinucleotide phosphate (NADPH) synthesis. 

Comments 2: Line 10-15: Add a sentence linking TROL-FNR’s role to climate resilience, e.g., "As drought stress intensifies globally, understanding redox regulation mechanisms is critical for engineering stress-tolerant crops."

Respond 2: Sentence is added and in revised manuscript is (lines 15-18, page 1, Abstract) Improvement in the knowledge of redox regulation mechanisms is critical for engineering stress-tolerant plants in the times of elevated global drought intensity. 

Comments 3: Line 13-14: “Our prior work indicates the TROL-FNR complex maintains redox equilibrium in chloroplasts and systemically in plant cells.”

Respond 3: Sentence is rewritten and in revised manuscript is (lines 13-16, page 1, Abstract) Our prior work indicates the TROL-FNR complex maintains redox equilibrium in chloroplasts and systemically in plant cells. 

Comments 4: Line 17: ARHO is used without defining it as an RHO-domain deletion mutant. Correct to "TROL ARHO (RHO-domain deletion mutant)" at first mention.

Respond 4: ΔRHO is defined as RHO domain deletion mutant in revised manuscript is (line 20, page 1, Abstract) TROL ΔRHO (RHO-domain deletion mutant).

Comments 5: Line 20: Replace vague descriptors (e.g., "successful scavenging") with quantitative trends: "TROL KO reduced O₂˙ˉ by …%, while ARHO reduced H₂O₂ by …%"

Respond 5: Vague descriptions are replaced with quantitative trends and in revised manuscript are (lines 23-25, page 1, Abstract) Under the drought stress conditions, TROL KO line showed ≈32% less O2˙ˉ, while TROL ΔRHO line showed ≈49% less H2O2 in comparison with the WT. 

Comments 6: Line 23: Explicitly state how findings advance the field: e.g., "This confirms TROL-FNR’s role as a master switch redirecting electrons to alternative sinks under stress, offering targets for improving crop resilience"

Respond 6: Advanced in the field are explicitly stated and in revised manuscript are (lines 25-28, page1, Abstract) This research confirms the  role of dynamical TROL-FNR complex formation in redox equilibrium maintenance by redirecting electrons in alternative sinks under stress, and also points it out as promising target for stress-tolerant plants engineering.

Comment 7: Line 54-59: Expand on RHO’s redox-sensing mechanism: "Inactive RHO may relay luminal redox cues via conformational changes in ITEP."

Respond 7: RHOs sensing mechanism is explained in more detail and in revised manuscript is (lines 64-81, page 2, Introduction) It has been postulated that inactive RHO domains participate in cooperation with MAPK phosphatases (a large group of signaling, regulatory enzymes), thioredoxins (TRX) and various stress response proteins. These domains are implicated in the maintenance of redox homeostasis and the detoxification of ROS across different intracellular processes. This implies a role for inactive RHO domains in signal transduction and cellular regulation. In A. thaliana, rhodanases activity is elevated under stress conditions, and stomata opening is at least partly regulated by receptors containing inactive RHO domains. Regulation of TROL-FNR binding is important for balancing redox status of stroma with the membrane ETC. Such regulation is critical for preventing over-reduction in these compartments and maintenance redox poise. Under stress conditions, it is hypothesized that FNR release from TROL could be triggered by signals originating from the RHO domain upon interaction with specific signaling molecules, leading to conformational changes in the ITEP domain. NMR structural analysis of the TROL RHO domain has revealed a slightly altered loop that may accommodate plastoquinone (PQ) as signal molecule. Additionally, progression of the eucaryotic cell cycle, controlled by Cdc25 phosphatase (active rhodanase domain), is influenced by intracellular redox changes. Quinone binding has been shown to modulate its activity, further highlighting the redox-sensitive regulatory functions of rhodanase domains [5, 10].

Comments 8: Line 68-71: Rewrite: “Electron excess in PETC disrupts energy balance, manifesting as: (1) LEF saturation via overoxidation, or (2) ROS overproduction via overreduction-induced photooxidation.”

Respond 8: Sentence is rewritten and in revised manuscript is (lines 90-95, page 3, Introduction) Electron excess  in PETC disrupts energy imbalance, manifesting as: (1)LEF saturation via overoxidation, or (2) ROS overproduction via overreduction-induced photooxidation. 

Comments 9: Line 189: Connect to climate change: “Drought disrupts CO₂ intake, causing PETC overreduction—a key agricultural challenge”

Respond 9: Sentence that introduces inclusion of drought stress is added in revised manuscript in (lines 201-202, page 6, Introduction) Since drought disrupts CO2 intake, the consequences are overreduction of PETC and restrictions in plant growth and development, which is significant agricultural obstruction. 

Comments 10: Line 198: Add references

Respond 10: Reference is added and is in revised manuscript is (line 213, page 7, Introduction) Moreover, detecting ROS at specific sites and under precise conditions while determining their origin remains a significant challenge [29]. Duanghathaipornsuk, S,; Farrell, E.J.; Alba-Rubio, A.C.; Zelenay, P.; Kim, D.S. Detection technologies for reactive oxygen species: fluorescence and electrochemical methods and their applications. 2021, 11, 30.

Comments 11: Line 199-211: Explicitly state the unresolved question: "While TROL-FNR’s role in chloroplast redox balance is established, its impact on whole-cell ROS propagation remains unproven."

Respond 11: Unresolved question is stated in revised manuscript (lines 219-221, page 7, Introduction) In addition to that, although the role of TROL-FNR protein pair in chloroplast redox homeostasis is established, its influence on the whole-cell redox homeostasis is yet to be approved. 

Comments 12: Line 201-202: Rewrite: “This approach minimized light-induced ROS artifacts and overcame compartment-specific detection challenges.”

Respond 12: Sentence is rewritten and in revised manuscript in (lines 215-219, page 7, Introduction) This approach minimized light-induced ROS artifacts and overcame compartment-specific detection challenges. 

Comments 13: Incomplete drought characterization. No physiological data (e.g., Relative water content (RWC), stomatal conductance, proline content, relevant enzymes….) confirming drought stress severity. Add supplementary table with stress parameters.

Respond 13: Physiochemical differences between growing substrates under normal humidity and drought stress conditions are stated in the Supplementary table 1 (Physiochemical characteristics of growing substrate under a) normal humidity, b) drought stress) that is added in attached revised manuscript, please see the table for details of substrate differences.

Comments 14: Line 217-219: Rewrite: “ROS differences between mutants and WT under contrasting growth conditions are visualized in the green fields images (Figs. 2,4,6).”

Respond 14: Sentence is rewritten and in revised manuscript is (lines 238-239, page 7, Results) ROS differences between mutants and WT under contrasting growth conditions are visualized in the green fields images.

Comments 15: Line 321: It's clear the authors moved the Materials and Methods section after the Discussion section and forgot to change the citation number. I think the numbering here should start at 39.

Respond 15: Manuscript segments were differently sorted before submission process which is the reason why references were numbered in wrong order. The citations are now sorted in the order of appearing in the text.

Comments 16: Line 348-349, 361, 441, 466, 467, 479: Add references.

Respond 16: References are added and in revised manuscript are (line 371, page 15, Discussion) [44-47].

Yang, X.; Lu, M.; Wang, Y.; Wang, Y.; Liu, Z.; Chen, S. Response mechanism of plants to drought stress. 2021, 7, 50.

Qiao, M.; Hong, C.; Jiao, Y.; Hou, S.; Gao,H. Impacts of drought on photosynthesis in major food crops and the related mechanisms of plant responses to drought. 2024, 13, 1808.

Hald, S.; Nandha, B.; Gallois, P.; Johnson, G.N. Feedback regulation of photosynthetic electron transport by NADP (H) redox poise. Biophys. Acta, Bioenerg. 2008, 1777, 433-440.

Golding, A.J.; Johnson, G.N. Down-regulation of linear and activation of cyclic electron transport during drought. 2023, 218, 107-114.

(line 373, page 15, Discussion) [45, 46, 48-54]. 

Qiao, M.; Hong, C.; Jiao, Y.; Hou, S.; Gao,H. Impacts of drought on photosynthesis in major food crops and the related mechanisms of plant responses to drought. 2024, 13, 1808.

Hald, S.; Nandha, B.; Gallois, P.; Johnson, G.N. Feedback regulation of photosynthetic electron transport by NADP (H) redox poise. Biophys. Acta, Bioenerg. 2008, 1777, 433-440.

Sun, H.; Shi, Q.; Liu, N.Y.; Zhang, S.B.; Huang, W. Drought stress delays photosynthetic induction and accelerates photoinhibition under short-term fluctuating light in tomato. 2023, 196, 152-161.

Bashir, N.; Athar, H.U.R.; Kalaji, H.M.,; Wróbel, J.; Mahmood, S.; Zafar, Z.U.; Ashraf, M. Is photoprotection of PSII one of the key mechanisms for drought tolerance in maize?. J. Mol. Sci. 2021, 22, 13490.

Lima-Melo, Y.; Alencar, V.T.; Lobo, A.K.; Sousa, R.H.; Tikkanen, M.; Aro, E.M.; Gollan, P.J. Photoinhibition of photosystem I provides oxidative protection during imbalanced photosynthetic electron transport in Arabidopsis thaliana. Plant Sci. 2019, 10, 916.

Lima‐Melo, Y.;, Gollan, P.J.; Tikanen, M.; Silveira, J.A.; Aro, E.M. Consequences of photosystem‐I damage and repair on photosynthesis and carbon use in Arabidopsis thaliana. Plant J. 2019, 97, 1061-1072.

Wang, Z.; Li, G.;, Sun, H.; Ma, L.; Guo, Y.; Zhao, Z.; Mei. L. Effects of drought stress on photosynthesis and photosynthetic electron transport chain in young apple tree leaves. Open. 2018, 7, :035279.

Fulgosi, H.; Vojta, L. Tweaking photosynthesis: FNR-TROL interaction as potential target for crop fortification.  Plant Sci.2020, 11, 318.

Allen, J.F. Cyclic, pseudocyclic and noncyclic photophosphorylation: new links in the chain. Trends Plant Sci. 2003, 8, 15-19.

(line 468, page 17, Discussion) [25, 26, 63].

Exposito-Rodriguez, M.; Laissue, P.P.; Yvon-Durocher, G.; Smirnoff, N.; Mullineaux, P.M. Photosynthesis-dependent H2O2 transfer from chloroplasts to nuclei provides a high-light signalling mechanism. commun. 2017, 8, 49.

Foyer, C.H.; Hanke, G. ROS production and signalling in chloroplasts: cornerstones and evolving concepts. Plant J. 2022, 111, 642-661.

Smirnoff, N.; Arnaud, D. Hydrogen peroxide metabolism and functions in plants. New phytol. 2019, 221, 1197-1214.

(line 493, page 18, Discussion) [67-69].

Dogra, V.; Kim, C. Singlet oxygen metabolism: from genesis to signaling. Plant Sci. 2020, 10, 1640.

Laloi, C.; Havaux, M. Key players of singlet oxygen-induced cell death in plants. Plant Sci. 2015, 6, 39.

Triantaphylidès, C.; Havaux, M. Singlet oxygen in plants: production, detoxification and signaling. Trends Plant Sci. 2009, 14, 219-228.

(line 508, page 19, Discussion) [70]. 

Zhang, A.; Jiang, M.; Zhang, J.; Ding, H.; Xu, S.; Hu, X.; Tan, M. Nitric oxide induced by hydrogen peroxide mediates abscisic acid‐induced activation of the mitogen‐activated protein kinase cascade involved in antioxidant defense in maize leaves. New Phytol. 2007, 175, 36-50.

Comments 17: Line 372, 433: Add the citation number: ‘Vojta et al. [… …]’

Respond 17: Citation numbers are added, in revised manuscript (lines 394, page 16, Discussion, line 457 page 17, Discussion), [8, 22]

Comments 18: Line 432: Rewrite: “ΔRHO mutants efficiently reduce H₂O₂, potentially through enhanced TROL-tAPX complex formation [Ref. no. …], boosting antioxidant activity.”

Respond18: Sentence is rewritten and in revised manuscript is (lines 455-459, page 17, Discussion) ΔRHO mutants efficiently  reduce H2O2, potentially through enhanced TROL-tAPX complex formation [22], boosting antioxidant activity. In addition, as proposed in Vojta et al. [22], TROL most likely forms a complex with the tAPX that is an important antioxidant enzyme in the process of H2O2 scavenging. 

Comment 19: Line 512: The numbering of citations here should change according to the change that will be made in the Discussion section.

Respond 19: Manuscript segments were differently sorted before submission process which is the reason why references were numbered in wrong order. The citations are now sorted in the order of appearing in the text.

Comment 20: Line 512-513: Rewrite: “Plants were grown from seeds in controlled-environment chambers.”

Respond 20: Sentence is rewritten and in manuscript is (lines 541-542, page 19, Materials and methods) Plants were grown from seeds in controlled-environment chambers. 

Comment 21: Line 531: Nutrient solution pH affects ROS stability. Add Hoagland’s solution pH value.

Respond 21: Nutrient solution is added and in revised manuscript is (line 559, page 19, Materials and methods) pH 5.5.

Comments 22: Line 549: Mention the age of the plants and their morphological characteristics at that stage. Specify leaf developmental stage sampled.

Respond 22: Age, morphology, and developmental stage of sampled plants are stated in revised manuscript in (lines 578-583, page 20, Materials and methods) For the normal humidity conditions plants were 4, while for the drought stress conditions plants were 6 weeks old. Morphologically, under normal humidity conditions leaves were of healthy green color with bigger rosettes in mutant lines than in WT. Under drought stress TROL KO plants had bigger rosettes of green color, while WT displayed reddish color form accumulated anthocyanins. Morphological characteristics of TROL ΔRHO were in between TROL KO and WT. All sampled leaves were at the fully developed stage.

Comments 23: Line 555: Correct: “… according to Prasad et al. [14].

Respond 23: in revised manuscript corrected sentence part is (lines 589-590, page 20, Material and methods) ...according to Prasad et al. [14].

Comments 24: Line 588: Address biological replicates: e.g. "Biological replicate = leaf from independently grown plant (n=… plants/genotype or line)."

Respond 24: Biological and technical replicates are described and stated in revised manuscript (lines 624-627, page 21, Materials and methods) Biological replicate = leaf from independently grown and randomly selected plants (n = 3 plants/plant line). Technical replicate = every leaf was imaged on 3 different locations in different plain levels to the total count of 10 images.

Comments 25: Line 591-593: Delete: “5. Conclusions” and “This section is not mandatory ….”

Respond 25: These lines are accidentally remained in the text as a part of submission template and are deleted in revised manuscript (lines 627-629, page 21, Materials and methods).

Comments 26: Line 604: Acknowledge limitations: Add: "While confocal imaging captured spatial ROS patterns, biochemical assays are needed to confirm scavenging kinetics.".

Respond 26: Limitations are acknowledged in the Conclusion of revised manuscript by adding sentence While confocal imaging captured spatial ROS patterns, biochemical assays are needed to confirm scavenging kinetics. (lines640-641, page 21, Conclusion)

Comments 27: Line 604: Reframe agricultural implications: e.g., "TROL manipulation shows potential for enhancing crop resilience pending field validation."

Respond 27:Agricultural implications are reframed in Conclusion of revised manuscript by adding sentence In TROL protein, agricultural improvements can find promising manipulation toll for improving crop resilience after validation in natural conditions. (lines 641-643, page 21, Conclusion)

Comments 28: The reference list must be formatted according to the journal's requirements.

Respond 27: The reference list is formatted according to the journals requirements and is in revised manuscript (pages 21-25, References).
